# Maturation State-Specific Alternative Splicing in *FLT3*-ITD and NPM1 Mutated AML

**DOI:** 10.3390/cancers13163929

**Published:** 2021-08-04

**Authors:** Anna Wojtuszkiewicz, Inge van der Werf, Stephan Hutter, Wencke Walter, Constance Baer, Wolfgang Kern, Jeroen J. W. M. Janssen, Gert J. Ossenkoppele, Claudia Haferlach, Jacqueline Cloos, Torsten Haferlach

**Affiliations:** 1Department of Hematology, Amsterdam University Medical Center, VU University Medical Center, Cancer Center Amsterdam, 1081 HV Amsterdam, The Netherlands; a.wojtuszkiewicz@amsterdaumumc.nl (A.W.); i.vanderwerf@amsterdamumc.nl (I.v.d.W.); j.janssen@amsterdamumc.nl (J.J.W.M.J.); g.ossenkoppele@amsterdamumc.nl (G.J.O.); 2MLL Munich Leukemia Laboratory, 81377 Munich, Germany; stephan.hutter@mll.com (S.H.); Wencke.Walter@MLL.com (W.W.); Constance.Baer@mll.com (C.B.); wolfgang.kern@mll.com (W.K.); claudia.haferlach@mll.com (C.H.); torsten.haferlach@mll.com (T.H.)

**Keywords:** alternative splicing, acute myeloid leukemia, *FLT3*-ITD, NPM1, FAB, maturation state

## Abstract

**Simple Summary:**

In hematological malignancies, genome-wide sequencing studies found the process of splicing to be surprisingly frequently disrupted. While recent studies characterized altered splicing in relation to splicing factor mutations in AML, this study explored differential splicing profiles associated with two most common aberrations in AML: *FLT3*-ITD and NPM1 mutations. We identified the differential splicing of FAB-type specific gene sets in *FLT3-ITD*+/NPM1+ specimens as compared to *FLT3*-ITD−/NPM1− samples. The primary functions perturbed by differential splicing in all three FAB types included cell cycle control and DNA damage response. Interestingly, differential expression mainly affected genes involved in hematopoietic differentiation. Our findings increase our understanding of how genetic mutations translate to phenotypic features of AML cells to further improve response predictions and to find innovative therapeutic approaches. Altogether, to the best of our knowledge, this is the first study to report differential splicing profiles associated with *FLT3*-ITD with a concomitant NPM1 mutation in AML.

**Abstract:**

Despite substantial progress achieved in unraveling the genetics of AML in the past decade, its treatment outcome has not substantially improved. Therefore, it is important to better understand how genetic mutations translate to phenotypic features of AML cells to further improve response predictions and to find innovative therapeutic approaches. In this respect, aberrant splicing is a crucial contributor to the pathogenesis of hematological malignancies. Thus far, altered splicing is well characterized in relation to splicing factor mutations in AML. However, splicing profiles associated with mutations in other genes remain largely unexplored. In this study, we explored differential splicing profiles associated with two of the most common aberrations in AML: *FLT3*-ITD and NPM1 mutations. Using RNA-sequencing data of a total of 382 primary AML samples, we found that the co-occurrence of *FLT3*-ITD and mutated NPM1 is associated with differential splicing of FAB-type specific gene sets. Despite the FAB-type specificity of particular gene sets, the primary functions perturbed by differential splicing in all three FAB types include cell cycle control and DNA damage response. Interestingly, we observed functional divergence between alternatively spliced and differentially expressed genes in *FLT3*-ITD+/NPM1+ samples in all analyzed FAB types, with differential expression affecting genes involved in hematopoietic differentiation. Altogether, these observations indicate that concomitant *FLT3*-ITD and mutated NPM1 are associated with the maturation state-specific differential splicing of genes with potential oncogenic relevance.

## 1. Introduction

Whole genome profiling efforts in the last decade have defined the somatic mutational landscape of acute myeloid leukemia (AML) [1,2]. While our knowledge regarding the genetics of AML has largely increased, little progress has been made with respect to improvements in the AML treatment outcome. Therefore, there is an urgent need to further deepen our understanding of how different genetic mutations affect phenotypic features of AML cells in order to better predict their responses to current treatments as well as to invent novel therapeutic approaches. The genetic lesions involved in AML pathogenesis include aberrations in transcription factors, epigenetic regulators and signaling molecules, which collaborate to promote a block in differentiation, paralleled by enhanced survival, self-renewal and proliferation [1,3,4]. Interestingly, alternative pre-mRNA splicing (AS) is known to play a pivotal role in the regulation of all these processes [5,6,7].

Pre-mRNA splicing is a crucial step in gene expression, whereby the non-coding segments (introns) are excised and coding regions (exons) are joined together [8]. Tissue and organ development was documented to be driven by coordinated networks of AS events, which regulate various aspects of differentiation, including cell cycle progression, DNA damage repair and apoptosis. Accordingly [5,6,7,9,10,11,12], AS perturbations were shown to affect these key facets of development, thereby facilitating oncogenesis [13,14]. In AML cells, it was estimated that almost 30% of expressed genes are aberrantly spliced as compared to non-malignant CD34+ progenitor cells [15]. Thus far, much attention has been dedicated to the characterization of AS in AML samples carrying mutations in splicing factors (SF); however, AS in relation to mutations in other genes, indirectly linked to splicing regulation, remains poorly characterized [16,17,18,19].

Internal tandem duplications (ITD) in the Fms-like tyrosine kinase 3 (*FLT3*) gene are among the most common genetic aberrations in AML, affecting roughly 30% of patients [1,2,20]. *FLT3* is a receptor tyrosine kinase, which, via downstream signaling pathways, controls the growth and survival of myeloid progenitors and is rendered constitutively active upon ITD insertion [21]. *FLT3*-ITD rarely occurs alone and most frequently coincides with mutations in nucleophosmin (NPM1) with many *FLT3*-ITD+/NPM1+ AML patients eventually relapsing [1]. NPM1 is a multifunctional protein with diverse physiological roles that include regulation of the cell cycle, DNA damage repair, maintenance of genomic stability and stress response [22]. The molecular synergy between *FLT3*-ITD and NPM1 mutations was demonstrated to drive rapidly developing AML in mouse models [23,24]. In contrast, co-occurring NPM1 and *NRAS* mutations induced less aggressive AML, underscoring the frequent occurrence and worse prognosis of *FLT3*-ITD+/NPM1+ AML [23,24]. This was paralleled by the cooperative impact of these two aberrations on gene expression profiles [23,25]. Interestingly, both *FLT3*-ITD and NPM1 were also previously shown to shape the epigenome in AML [22,26]. As the process of splicing occurs co-transcriptionally and its regulation was shown to be influenced by chromatin status (including modifications to both histones and DNA), by shaping the epigenome *FLT3*-ITD and NPM1 mutations have the potential to affect splicing [27,28,29]. Yet splicing profiles associated with these co-occurring aberrations have not been studied thus far.

Therefore, the aim of this study was to explore the differential splicing profiles associated with the presence of *FLT3*-ITD with concomitant NPM1 mutations to characterize their potential oncogenic relevance. Furthermore, AS profiles as well as gene expression profiles orchestrate differentiation and maturation of cells and tissues and therefore, can show much variability between various cell types and maturation stadia [30,31]. Since both *FLT3*-ITD and NPM1 mutations occur in AML cells arrested in different maturation stadia, we evaluated whether differential splicing and differential expression signatures in relation to *FLT3*-ITD and NPM1 mutations in AML showed FAB subtype specificity.

## 2. Materials and Methods

### 2.1. Patient Samples

For the splicing analysis, 382 untreated bone marrow (BM) and peripheral blood (PB) samples collected at the time of diagnosis from AML patients were used. This included 327 samples in the discovery cohort (collected at MLL Munich Leukemia Laboratory, Munich, Germany) and 55 samples in an independent validation cohort (collected at Amsterdam University Medical Center, AUMC, location VUmc). All patients signed a written informed consent. The cell type-specific analyses included the three most represented subtypes in the dataset, according to the French–American–British (FAB) classification (72 M1, 92 M2 and 56 M4 samples). The validation cohort included 19 M1, 17 M2 and 19 M4 specimens (sample metadata are listed in Appendix A). This study was approved by the internal review board of the MLL and local ethics committee of Amsterdam UMC and was conducted in accordance with the Declaration of Helsinki.

### 2.2. Genetic Profile

The presence of *FLT3*-ITD, NPM1 mutations and SF mutations of patients from the MLL dataset were based on routine molecular diagnostics (including a combination of gene scan analysis, melting curve analysis, Sanger sequencing and next-generation amplicon sequencing as described previously) complemented by whole genome sequencing (see Supplemental Methods) [32,33,34,35]. The mutational status in the AUMC dataset was defined based on the molecular diagnostics as described previously, complemented with variant calling from RNA sequencing data (Supplemental Methods) [36,37]. All samples carrying SF mutations (*SF1, SF3A1, SF3B1, SRSF2, U2AF1, U2AF2* and *ZRSR2*) or samples for which the average coverage in frequently mutated exons of SF genes was low were removed from the analysis (see Supplemental Methods). All samples considered *FLT3*-ITD+ based on the mutational analysis were included in the analysis (including the following fractions of samples with *FLT3*-ITD allelic ratio > 0.5 as determined on DNA by fragment analysis: 63.6% of M1, 47.1% of M2 and 58.3% of M4 samples).

### 2.3. RNA Sequencing

Total RNA was extracted from BM and PB samples using the MagNA Pure 96 Instrument and the MagNA Pure 96 Cellular RNA LV Kit (Roche LifeScience, Mannheim, Germany) for the discovery cohort, and using the RNeasy mini kit (Qiagen, Venlo, the Netherlands) for validation cohort. The TruSeq Total Stranded RNA kit was used to generate RNA libraries following the manufacturer’s recommendations, starting with 250 ng of total RNA (Illumina, San Diego, CA, U.S.A.). The 2 × 100 bp paired-end reads were sequenced on the NovaSeq 6000 with a median of 50 mln reads per sample (Illumina). Using BaseSpace’s RNA-seq Alignment app (v2.0.1) with default parameters, reads were mapped with the STAR aligner (v2.5.0a) to the human reference genome hg19 (RefSeq annotation). For gene expression analysis, estimated gene counts were normalized applying Trimmed mean of M-values (TMM) normalization method of the edgeR package [38]. The resulting log2 counts per million (CPMs) were used as a proxy of gene expression. Genes with a CPM < 1 were filtered out.

### 2.4. Differential Gene Expression and Splicing Analysis

Gene expression differences were assessed using the limma package [39] with false discovery rate (FDR) correction for multiple testing. Genes with an FDR less than 0.05 and an absolute logFC greater than 1.5 were considered differentially expressed (DE).

rMATS version 4.0.2 was used to detect alternative splicing (AS) events [40]. rMATS is able to quantify four major types of alternative splicing events: skipped exons (SE), alternative 5′ splice site selection (A5SS), alternative 3′ splice site selection (A3SS) and retained introns (RI). The difference in splicing between the two groups is expressed as ΔPSI (proportion spliced-in). AS events supported by fewer than 10 counts per sample were filtered out. AS events were considered significantly differential when FDR < 0.05 and absolute ΔPSI > 0.1. The Z-score calculation and hierarchical clustering for AS events were performed using PSI values. The data were visualized using ggplot2 [41] and ComplexHeatmap packages (version 2.2.0) [42] in R (versions 3.5.3 and 3.6.2). Protein domains directly affected by splicing events were determined using the Maser package (version 1.0.0) [43] in R (versions 3.5.3) upon conversion of the genomic coordinates from hg19 to hg38 assembly (using AnnotationHub v. 2.14.5, GenomicRanges v. 1.34.0 and rtracklayer v. 1.42.2 packages in R) [44,45,46]. Motif enrichment analysis for the differentially spliced splicing factors was performed using rMAPS tool [47,48]. For gene ontology analysis, gene IDs for significant AS events were uploaded into the STRING tool (v11.0) to retrieve interactions [49]. STRING interaction networks were imported and annotated in Cytoscape (v3.8.1) [50]. Gene ontology analysis was performed within Cytoscape using the ClueGO plugin [51].

For validation of AS events in an independent sample cohort, AS events and their respective PSI values in the validation cohort were determined by rMATS. Subsequently, the PSI values in the validation cohort, corresponding to significant AS events in the discovery cohort, were retrieved based on genomic coordinates and compared between *FLT3*-ITD+/NPM1+ and *FLT3*-ITD−/NPM1− sample groups, using the Mann–Whitney U test. The raw results of differential splicing analyses are available in Appendix A, while the lists of differentially expressed genes are available in Appendix A.

## 3. Results

### 3.1. Alternative Splicing Profiles of FLT3-ITD and NPM1 Double Mutated Cells Show High FAB-Type Specificity

*FLT3*-ITD and NPM1 mutations were previously reported to be associated with specific differential gene expression profiles in AML (Appendix A) [25,52]; however the influence of these aberrations on splicing has not been studied thus far. Therefore, we applied the rMATS algorithm (see methods) to analyze differential splicing in relation to the presence of these mutations in an RNA-sequencing dataset obtained from 327 diagnosis samples of de novo AML patients (with patients carrying splicing factor mutations excluded from the analysis to avoid bias). Since co-occurrence of *FLT3*-ITD and NPM1 mutations was previously described to exert particularly strong synergistic effects on gene expression [23,24,53], we primarily focused our splicing analysis on this double mutated subset of the samples. This approach uncovered a total of 217 significant differential splicing events in *FLT3*-ITD+/NPM1+ specimens as compared to *FLT3*-ITD−/NPM1− samples (Appendix A). Hierarchical clustering did not reveal a specific cluster of *FLT3*-ITD+/NPM1+ samples, suggesting limited specificity of the identified splicing events for this subgroup (Appendix A).

Since cell type and the maturation state of the cells are known to influence alternative splicing [5,30,54,55,56,57], we stratified the cohort on the basis of individual (most common) FAB types, including 48 M1, 80 M2 and 30 M4 samples (Appendix A). Interestingly, within specific FAB types, we could identify patterns of differential splicing more specific for *FLT3*-ITD+/NPM1+ samples, suggesting highly maturation state-dependent splicing regulation in the context of these mutations (Figure 1A,B). The majority of significant events in all analyses constituted skipped exons (Figure 1C, Appendix A).

Furthermore, the splicing profiles of the double mutated *FLT3*-ITD+/NPM1+ samples showed an improved clustering pattern as compared to either *FLT3*-ITD or NPM1, overall (Figure 1, Appendix A). This points to a possible synergy in the splicing regulation between *FLT3*-ITD and NPM1 mutations, similar to that observed for regulation of gene expression. Strikingly, the number of differentially spliced genes in *FLT3*-ITD+/NPM1+ samples was remarkably high in the M4 subtype (1438 differential splicing events) as compared to the M1 and M2 samples (approximately 200 events each, Figure 1A). In addition, the overlap between differentially spliced genes in *FLT3*-ITD+/NPM1+ samples in the three individual FAB types encompassed only 12 genes (Figure 1D), highlighting the impact of FAB type on differential splicing profiles associated with these genetic aberrations. The functional annotation of these genes does not give direct clues as to the relation with *FLT3* or NPM1 but are more related to splicing and regulation of protein homeostasis. Interestingly, similar to splicing events, differentially expressed genes in *FLT3*-ITD+/NPM1+ samples were largely FAB type-specific (Appendix A), including several regulators of hematopoietic differentiation previously reported to be associated with *FLT3*-ITD and NPM1 mutations (i.e., differential expression of FOXC1, MEIS1 and FOXO1 in M1 and M2 but not in M4 samples, Appendix A). Altogether, these findings demonstrate that differentially spliced (as well as aberrantly expressed) genes associated with *FLT3*-ITD and mutated NPM1 might be relevant only in AML cells of specific differentiation stages.

### 3.2. FLT3-ITD and NPM1 Double Mutated Cells Display Altered Splicing of Genes Involved in Cell Cycle Control, DNA Damage Response and Signaling Pathways

To evaluate the biological functions of the uncovered differential splicing events in *FLT3*-ITD+/NPM1+ samples, we performed functional enrichment analysis. Remarkably, in the three FAB types M1, M2 and M4, the major affected processes included regulation of the cell cycle and DNA damage repair (26 genes in M1, 29 genes in M2 and 171 genes in M4 subtype, Figure 2), although the particular repertoires of genes implicated in these functions were FAB type-specific.

This included, for instance, two components of the BRCA1-A complex (BABAM1 and BABAM2/BRE) and CEP164, an ATR/ATM signaling regulator in the M1 samples, and two genes coding for centromeric proteins (CENPE and CENPJ) as well as PLK4, a kinase that plays a central role in centriole duplication in M2 specimens. Factors controlling DNA damage response and cell cycle constituted a large network among the numerous, differentially spliced genes found in *FLT3*-ITD+/NPM1+ M4 samples. This network included, for instance, genes with an established role in oncogenesis, such as ATR, BRCA2, TOP2A, TOP2B, and the Aurora kinases (AURKA and AURKB), as well as the MELK kinase, an important regulator involved in both the cell cycle control, self-renewal and apoptosis.

Next to the cell cycle control and DNA damage repair, several genes differentially spliced in relation to *FLT3*-ITD and NPM1 mutations were involved in signaling pathways that regulate survival and proliferation of AML cells (Figure 3). In M1 patients, which is the most undifferentiated of the three analyzed FAB types, the network of significantly differentially spliced genes in *FLT3*-ITD+/NPM1+ samples included EZH2, an important regulator of hematopoietic stem cells, as well as two genes that regulate development of embryonic stem cells (RBBP5 and JARID2). The most pronounced perturbation of signaling in (the more mature) *FLT3*-ITD+/NPM1+ M2 specimens involved NOTCH signaling (FBXW1, RBX1, JAG1, NCOR2 and HDAC6, Figure 3B) and apoptosis regulation (i.e., NME4 and APIP). The *FLT3*-ITD+/NPM1+ samples of M4 FAB type displayed differential splicing of many factors involved in survival signaling (the entire network in Figure 3C). Prominent examples include genes coding for subunits of phosphoinositide 3-kinase (PI3K), including three catalytic subunits (PIK3CA, PIK3CB and PIK3CG) and one regulatory subunit (PIK3R5). Remarkably, the PI3K/AKT signaling pathway was found to be perturbed by differential expression in M1 and M2 FAB types but not in M4 specimens.

Interestingly, overall, we found very little overlap between differentially spliced and differentially expressed genes in relation to concomitant *FLT3*-ITD and NPM1 mutations (Appendix A). While both types of regulation affected genes involved in various survival signaling pathways, the major processes regulated by differential expression and splicing varied with differentially expressed genes primarily implicated in hematopoietic differentiation (i.e., HOX genes, FOXC1, MEIS1 and FOXO1, Appendix A). In summary, differential splicing in *FLT3*-ITD+/NPM1+ cells perturbed regulators of processes highly relevant for oncogenesis, including progression through the cell cycle and DNA damage response as well as survival signaling. Furthermore, these two types of gene expression regulation (differential expression and splicing) appear to complement each other in the two important aspects of oncogenesis: uncontrolled proliferation and impaired differentiation.

### 3.3. Factors Regulating Differential Splicing in the Context of FLT3-ITD and NPM1 Mutations

The differential splicing profiles found in *FLT3*-ITD+/NPM1+ samples could be the result of differential expression of splicing factors in this context. We did not find any splicing regulators to be differentially expressed in *FLT3*-ITD+/NPM+ samples in none of the FAB types. However, since splicing factors are known to autoregulate their own splicing, we also looked at differentially spliced splicing regulators (Figure 4 and Appendix A). In the M1 subtype, only the CELF2 splicing factor was found to be differentially spliced in *FLT3*-ITD+/NPM1+ specimens. Notably, *FLT3*-ITD+/NPM1+ M2 samples displayed altered splicing of 9 splicing regulators (CELF2, RBM38, RBM39, DDX16, PUM1, SRSF10, PRMT7, ZRANB2 and TFIP11), and 18 splicing factors were differentially spliced in *FLT3*-ITD+/NPM1+ M4 specimens (i.e., CLK2, SRPK1, SRSF10, HNRNPC, HNRNPLL, PTBP1, RBM3 and RBM5).

To evaluate if these specific splicing factors possibly contributed to the splicing regulation in the investigated sample set, we determined whether sequences (motifs) recognized and bound by these splicing regulators were enriched in the proximity to the significant differential splicing events, as compared to non-differentially spliced exons (using rMAPS tool; Figure 4 and Appendix A). Remarkably, CELF2 motifs were enriched in differential splicing events in *FLT3*-ITD+/NPM+ M1 and M2 samples. In M2, we also detected enrichment of SRSF10, RBM38 and PUM1 motifs. Similarly, the differential splicing events identified in *FLT3*-ITD+/NPM+ M4 specimens were enriched for SRSF10, HNRNPC, PTBP1, RBM3, RBM5 and HNRNPLL motifs. Overall, these data indicate that differentially spliced splicing factors found in the current analysis are likely to at least partly contribute to the global changes in splicing profiles in the context of co-occurring *FLT3*-ITD and NPM1 mutations in the three FAB types.

### 3.4. Evaluation of the Relevance of Differential Splicing Events

To gain more insight into the relevance of the identified splicing events, we next tested if they are likely to alter the function of the resultant proteins. To address this question, we evaluated whether sequences coding for functional protein domains were directly affected by selected splicing events using, the Maser tool (Appendix A). In this analysis, we focused on genes involved in the cell cycle, DNA repair and cell signaling, specifically. Interestingly, the vast majority of AS events found in the *FLT3*-ITD+/NPM1+ samples in all three FAB types were predicted to directly alter functional protein domains (M1: 83.6%, M2: 92.5%, M4: 86.2%) and are, therefore, likely to change or even abrogate the function of their corresponding proteins.

Finally, to further substantiate our findings, differentially spliced genes were validated in an independent sample set, which included 19 M1 samples, 17 M2 samples and 19 M4 specimens. We found that 33.3% of differentially spliced genes in *FLT3*-ITD+/NPM1+ M1 samples showed a tendency (*p*-value < 0.2) toward differential splicing in the validation set (Appendix A). The same was true for 21.7% of genes in the M2 subset (Appendix A) and 28.7% of M4 specimens (Appendix A and selected examples that showed the same trend in discovery and validation cohorts in Figure 5 and Appendix A). Taken together, while our analysis indicates large heterogeneity in splicing between samples, it does support the relevance of many of the identified differential splicing events in the genes involved in regulation of the cell cycle and DNA damage repair as well as signaling.

## 4. Discussion

To the best of our knowledge, this is the first study to report differential splicing profiles associated with *FLT3*-ITD with a concomitant NPM1 mutation in AML. It was previously demonstrated that *FLT3*-ITD collaborates with NPM1 mutations in regulating chromatin state and gene expression profiles to drive AML [23,24,25,52,53]. Our data suggest that this cooperative regulation is further extended to alternative splicing. Importantly, we found that there appears to be no universal splicing profile associated with concomitant *FLT3*-ITD and NPM1 mutations that would transcend all subtypes of AML cells. Instead, we find that the co-occurrence of these two aberrations is associated with differential splicing of FAB subtype-specific sets of genes. This is in line with the crucial role of alternative splicing in the differentiation of cells, including hematopoiesis [22,23,52]. While the FAB-type specificity was very pronounced for differential splicing profiles, it also affected differential gene expression profiles. For instance, the upregulation of *MEIS1* and *FOXC1*, previously reported to be associated with mutated NPM1 and to regulate stem-like properties [31], was only noted in the *FLT3*-ITD+/NPM1+ samples of M1 and M2 FAB types, but not in M4 specimens. These observations indicate that the relevance of differential splicing and expression of important contributors to leukemogenesis is limited to certain differentiation stages of AML. Accordingly, a recent study reported that specific subsets of differentially expressed genes associated with relapse in AML only have prognostic value within specific molecular subsets (i.e., *MLL* rearranged) and FAB types [22].

Interestingly, FAB type-specific differentially spliced genes were primarily involved in cell cycle control and DNA damage response, suggesting that perturbation of different genes involved in the same process could give a similar outcome (i.e., deregulation of the cell cycle). The normal physiological functions of NPM1 include maintenance of genomic stability by regulation of DNA repair and cell cycle progression [4,22,23]. Accordingly, mutated NPM1 was previously linked to increased genomic instability and subsequent acquisition of additional mutations that activate signaling pathways (i.e., STAT or RAS) [58]. Since NPM1 mutations are thought to occur before *FLT3*-ITD, it is conceivable that the differential splicing of genes involved in cell cycle regulation and DNA damage repair that we observed in *FLT3*-ITD+/NPM1+ samples arose due to NPM1 mutations, or upon additional subsequent changes. The splicing perturbation of these processes could contribute to genomic instability, thereby facilitating acquisition of *FLT3*-ITD.

Strikingly, we noted functional divergence between differentially spliced and differentially expressed genes. While the first type of regulation primarily perturbed genes involved in cell cycle control and DNA damage response, the latter affected genes involved in hematopoietic differentiation. Since both of these processes constitute crucial and complementary aspects of oncogenesis, it appears that regulation at the level of gene expression and alternative splicing complement each other to drive the development of AML. Finally, the extent of differential splicing in relation to *FLT3*-ITD and NPM1 mutations in M4 FAB type was particularly large.

This could be partly related to the larger diversity of cells classified into this FAB type, which includes next to promyelocytes and more mature cells of the granulocytic lineage, as well as more than 20% of cells with monocytic features [5]. FAB subtypes were used in the current study as an approximation of specific maturation stadia of AML cells. While this classification was useful for our pilot analysis, subsequent studies should examine the maturation state specificity of differential expression and splicing in various purified immunophenotypic (and molecularly defined) subtypes of AML cells.

To a great extent, the detected splicing events were predicted to directly affect the functional protein domains and therefore, are likely to have an impact on the phenotype of AML cells. This should be further confirmed in functional studies. In addition, we found that 21.7–33.3% of differential splicing events from the discovery set showed a similar trend in an independent validation sample set. Although this analysis validated many events, it also suggests a relatively large heterogeneity in splicing between AML samples. Despite the initially large number of AML patient specimens in the current study, the sample numbers in the subtype analyses were substantially lower.

Therefore, splicing profiles of *FLT3*-ITD+/NPM1+ cells should be further confirmed in larger datasets. This would also allow the assessment of differential splicing in *FLT3*-ITD+ samples without mutated NPM1 (and vice versa), as well as focusing on *FLT3*-ITD+ samples with a high allelic ratio.

Finally, development and differentiation-related coordinated networks of alternative splicing events were previously reported to be orchestrated by RNA-binding proteins [27,28,29]. Although we did not find any differentially expressed splicing regulators in our dataset, we found differentially spliced splicing factors in each FAB subtype, for which binding motifs were enriched in the identified differential splicing events. Therefore, these regulators are likely to, at least partly, contribute to the differential splicing profiles of *FLT3*-ITD+/NPM1+ AML samples. Future studies should confirm the binding of these specific splicing factors in the vicinity of alternative splicing events. As the process of splicing occurs predominantly co-transcriptionally, its regulation is tightly coupled to the transcription and chromatin status, including modifications to both histones and DNA [59,60]. Recent studies showed that mutated *IDH2* as well as *RUNX1* knockout alter the splicing profiles [61]. Furthermore, dynamic changes in histone modifications were shown to predominantly occur in exons that were differentially spliced during differentiation of human embryonic stem cells, demonstrating that the chromatin status can directly affect splicing, thereby driving cell differentiation [61]. Since both *FLT3*-ITD and NPM1 mutations affect the chromatin status, they are also likely to indirectly influence splicing profiles through changes in histone and DNA modifications. This could not be explored in the current dataset due to the lack of data on histone and DNA modifications but should be further evaluated to fully elucidate the mechanisms behind the splicing regulation in the context of *FLT3*-ITD and mutated NPM1.

## 5. Conclusions

Altogether, these data show that concomitant *FLT3*-ITD and NPM1 mutations are associated with FAB type-specific altered splicing of genes with potential relevance for oncogenesis. Subgroup specific splicing analysis, stratified on FAB subtypes, pointed to important features (especially related to splicing) of cells carrying the same genetic aberrations (i.e., *FLT3*-ITD+/NPM1+) but arrested in a different stage of differentiation. Although FAB classification is not of prognostic use, maturation context-specific differential splicing analyses identified genes involved in critical cellular processes, including regulation of DNA damage, and survival signaling. Based on the functional relevance of such genes, alternative splicing could potentially affect the response of cells to (chemo- and targeted) therapy. Interesting examples of genes that could guide the selection of cell type-specific therapeutic targets include EZH2 in *FLT3*-ITD+/NPM1+ M1 samples, Notch signaling in M2 patients and PI3K/AKT signaling or MELK kinase in M4 subtype. However, future studies should further explore the functional relevance of cell type-specific differential splicing in *FLT3*-ITD+/NPM1+ AML cells in order to determine their impact on response to treatment and usefulness as novel therapeutic targets.

## Figures and Tables

**Figure 1 cancers-13-03929-f001:**
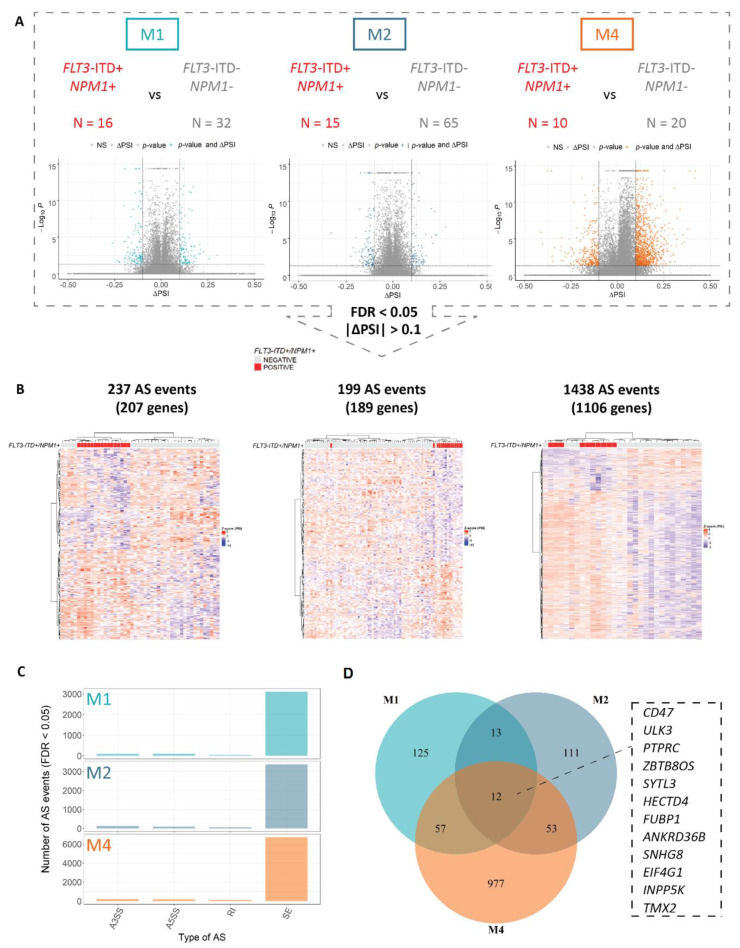
Concomitant *FLT3*-ITD and NPM1 mutations are associated with a strong FAB type-specific splicing profile. An overview of the differential splicing analysis performed with rMATS in three major FAB subtypes of AML (M1, M2 and M4) including (**A**) volcano plots of PSI (proportion spliced-in) values and (**B**) hierarchical clustering performed using significant differential splicing events (FDR < 0.05) with a minimal splicing difference between the two groups of 0.1 (|ΔPSI| > 0.1). The numbers of *FLT3*-ITD+/NPM1+ and *FLT3*-ITD−/NPM1− patients as well as the number of significant differential splicing (AS) events in each analysis are indicated. (**C**) The distribution of significant AS events in M1, M2 and M4 FAB types between the four main AS categories—skipped exons (SE), alternative 3′ splice site selection (A3SS), alternative 5′ splice site selection (A5SS) and retained introns (RI). (**D**) Overlap between significantly differentially spliced genes in relation to concomitant *FLT3*-ITD and NPM1 mutations in M1, M2 and M4 FAB subtypes with a minimal splicing difference between the two groups of 0.1 (FDR < 0.05, |ΔPSI| > 0.1). The 12 events overlapping between all three FAB subtypes are indicated.

**Figure 2 cancers-13-03929-f002:**
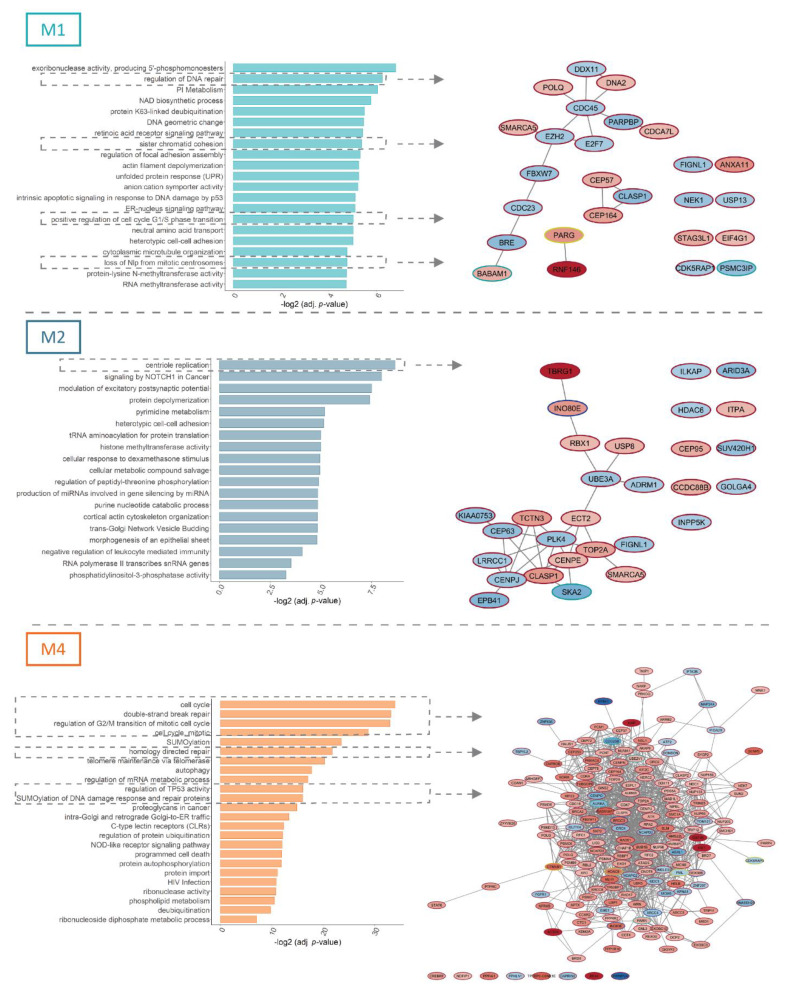
Functional analysis of differentially spliced genes in relation to concomitant *FLT3*-ITD and NPM1 mutations. The figure depicts functional enrichment among significant differential splicing events (FDR < 0.05) in relation to concomitant *FLT3*-ITD and NPM1 mutations in M1, M2 and M4 FAB subtypes with a minimal splicing difference between the two groups of 0.1 (|ΔPSI| > 0.1). For each FAB subtype, processes related to cell cycle control and DNA damage response are highlighted and depicted as protein networks. Node fill color signifies the ΔPSI value for each differential splicing event, while the color of the node edge codes for the type of differential splicing event: alternative 3′splice site selection (A3SS, blue), alternative 5′ splice site selection (A5SS, turquoise), retained intron (RI, yellow) and skipped exon (SE, brown).

**Figure 3 cancers-13-03929-f003:**
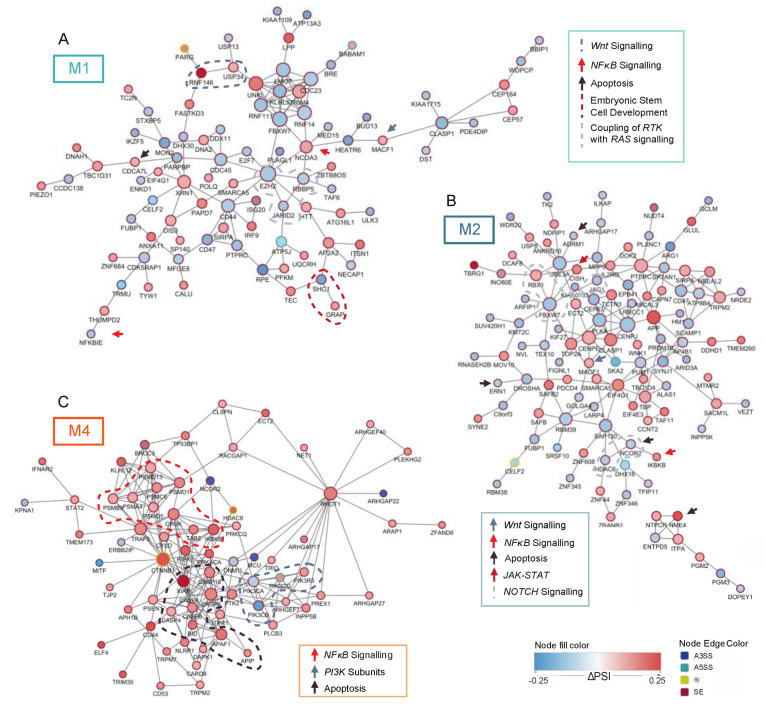
Networks of genes relevant to oncogenesis. (**A**) Network of all significantly differential splicing events with the minimal splicing difference between groups of 0.1 (FDR < 0.05, |ΔPSI| > 0.1) in relation to *FLT3*-ITD and mutated NPM1 in M1 patients. (**B**) Network of all significant differential splicing events with the minimal splicing difference between groups of 0.1 (FDR < 0.05, |ΔPSI| > 0.1) in relation to *FLT3*-ITD and mutated NPM1 in M2 patients. (**C**) Subnetwork of genes involved in signaling pathways significantly differentially spliced (FDR < 0.05, |ΔPSI| > 0.1) in relation to *FLT3*-ITD and mutated NPM1 in M4 patients. In each panel, genes relevant for particular signaling pathways are highlighted. Node size indicates connectivity of the genes. Node fill color signifies the ΔPSI value for each differential splicing event while the color of the node edge codes for the type of differential splicing event: alternative 3′splice site selection (A3SS, blue), alternative 5′ splice site selection (A5SS, turquoise), retained intron (RI, yellow) and skipped exon (SE, brown).

**Figure 4 cancers-13-03929-f004:**
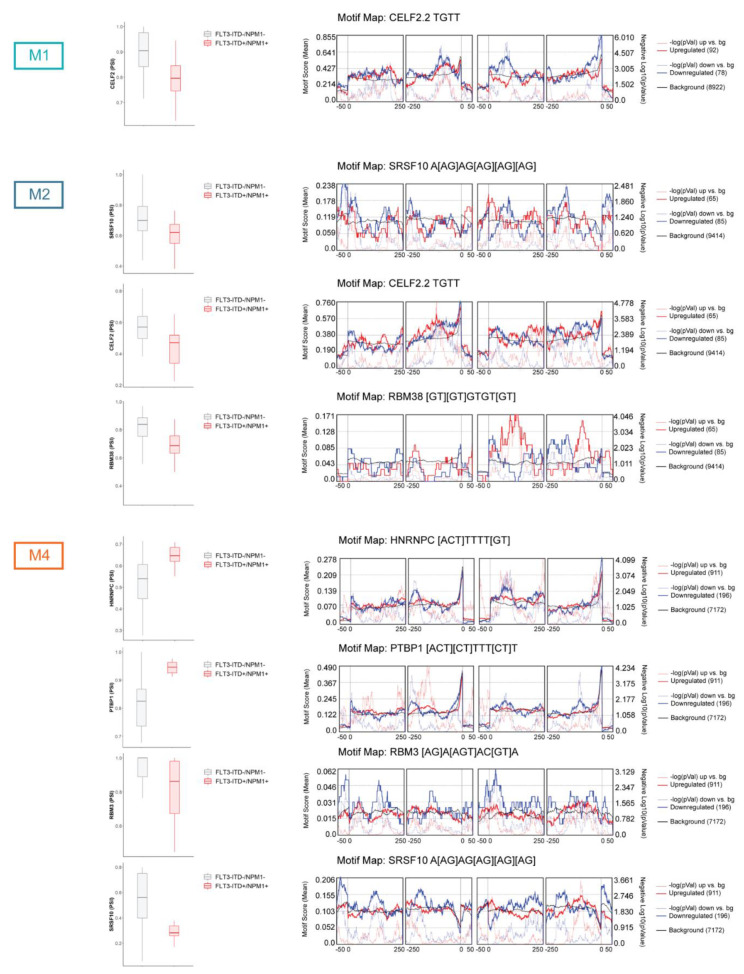
Differential splicing of splicing factors and motif enrichment analysis in relation to *FLT3*-ITD and NPM1 mutations. The figure depicts PSI values for selected differentially spliced splicing regulators (FDR < 0.05 and |ΔPSI| > 0.1) in M1, M2 and M4 FAB types as well as motif enrichment analysis for these splicing regulators performed by rMAPS2. This tool evaluates enrichment of motifs recognized by specific splicing factors in significantly differentially spliced events as compared to the background events (all splicing events detected by rMATS, including non-differential events). The motif enrichment is assessed in the differentially spliced exons as well as immediate upstream and downstream sequences. The motif enrichment score (the left y axis) is depicted by the solid blue (for events with ΔPSI < 0) and solid red (for events with ΔPSI > 0) lines. The solid black line indicates motif score for the background events. The negative logarithm of the *p*-value (right y axis) is depicted by the broken blue (for events with ΔPSI < 0) and red (for events with ΔPSI > 0) lines.

**Figure 5 cancers-13-03929-f005:**
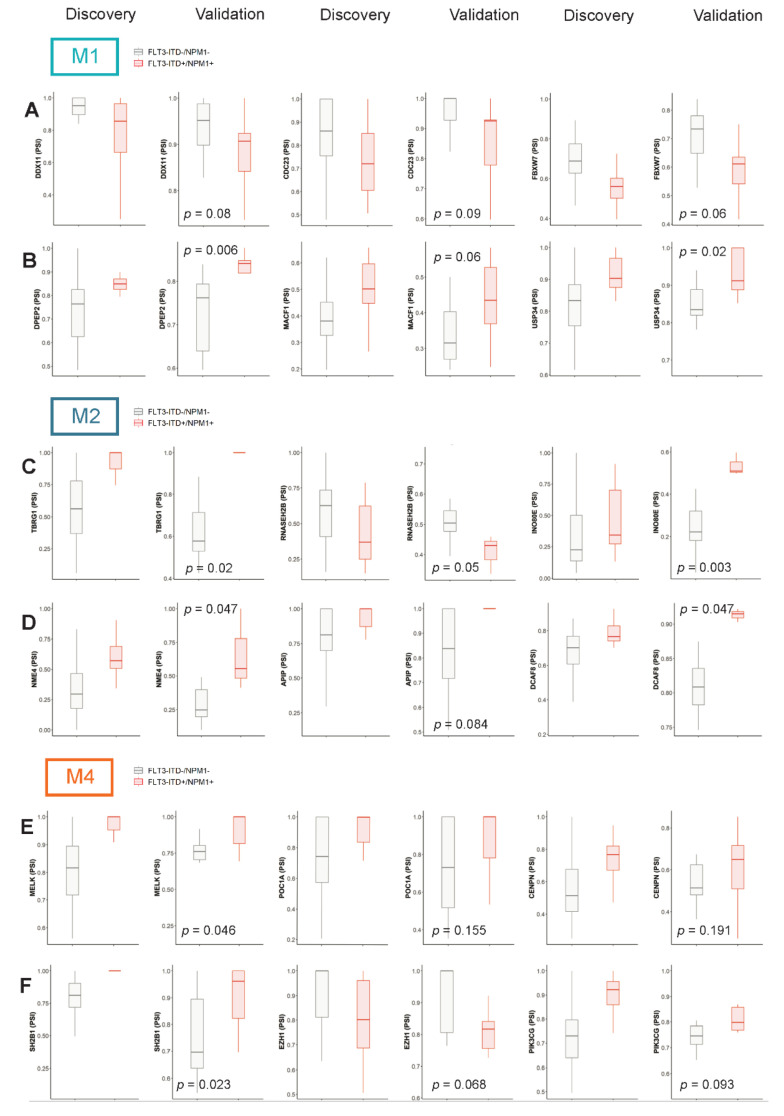
Validation of selected differential splicing events in an independent sample set. The figure depicts PSI values for selected differential splicing events in our initial discovery cohort (all events with FDR < 0.05 and |ΔPSI| > 0.1) and an independent validation cohort in M1, M2 and M4 FAB types. (**A**,**C**,**E**)—Selected splicing events affecting genes involved in cell cycle regulation and DNA damage response in M1 (*DDX11, CDC23, FBXW7*), M2 (*TBRG1, RNASEH2B, INO80E*) and M4 (*MELK, POC1A, CENPN*) FAB types, respectively. (**B**,**D**,**F**) Selected splicing events affecting genes involved in signaling pathways in M1 (*DPEP2, MACF1, USP34*), M2 (*NME4, APIP, PCAF8*) and M4 (*SH2B1, EZH1, PIK3CG*) FAB types, respectively.

## Data Availability

The raw data used during the current study are available from the corresponding author on reasonable request.

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
