# Peer review of "Maturation State-Specific Alternative Splicing in FLT3-ITD and NPM1 Mutated AML"

_cancers, 2021, doi:10.3390/cancers13163929_

Round 1

Reviewer 1 Report

This is innovative work by Wojtuszkiewicz et al. identifying differential splicing events in genes contributing to cell cycle, DNA damage repair and signalling in AML patients.

One major finding is a large heterogeneity between samples and no distinct cluster could be identified for the subgroup of interest consisting of FLT3-ITD and NPM1 co-mutated patients.

Instead, the data shows that concomitant FLT3-ITD and NPM1 mutations are associated with FAB type-specific altered splicing of genes.

The work is lacking functional experiments on potential therapeutic relevances of the findings.

Minor findings:

It would be helpful to better elucidate the genetic background of the samples (e.g. influcence of Co-Mutations of the AML samples on splicing clusters?);

Did the authors find any differences in FLT3-ITD allelic ratio, type of ITD (e.g. length, localization) regarding the genes involved altered splicing?

Author Response

The work is lacking functional experiments on potential therapeutic relevances of the findings.

Response: Thank you for acknowledging the innovative nature of our work. As we agree that our work is rather descriptive, we have slightly adjusted our results and conclusions in our manuscript to tone down our conclusion.

Minor findings

It would be helpful to better elucidate the genetic background of the samples (e.g. influcence of Co-Mutations of the AML samples on splicing clusters?);

Response: We determined the frequency of recurrent mutations, however, only DNMT3A mutation was present in a large enough part of the patients that allowed meaningful further analysis.  Supplemental Table S1 and S2 show the number of DNMT3A mutations. We have studied the influence of DNMT3A mutations carefully, but did not identify influences of the mutation on the clustering. Despite initially large number of AML patient specimens in the current study, the sample numbers in the subtype analyses were substantially lower. Therefore, potential effects  of co-mutations on differential splicing should be further analyzed in future larger datasets.

In addition, to avoid any influence of co-mutations related to splicing, all samples carrying SF mutations (SF1, SF3A1, SF3B1, SRSF2, U2AF1, U2AF2 and ZRSR2) or samples for which the average coverage in frequently mutated exons of SF genes was low, were removed from the patient cohort and from further analysis (Methods Line 120-122; Supplemental Methods).

Did the authors find any differences in FLT3-ITD allelic ratio, type of ITD (e.g. length, localization) regarding the genes involved altered splicing?

Response: We acknowledge the relevance of the allelic ratio and length of the ITD for its potential biological effect. We carefully studied the differences in FLT3-ITD allelic ratio in our dataset. However, we did not identify clear differences in clustering based on differences in allelic ratio as shown in Figure S2 and Figure S3. ITD length data were not available for all patients. Importantly, despite initially large number of AML patient specimens in the current study, the sample numbers in the subtype analyses were substantially lower. Therefore, differential splicing in FLT3-ITD+ samples with high allelic ratio or long ITD length, should be further studied in larger datasets.

Reviewer 2 Report

In the manuscript Dr Wojtuszkiewicz and colleagues aimed to delineate the differential splicing profiles associated with concurrent FLT3-ITD and NPM1 mutations in AML. They found that functional divergence between alternatively spliced and differentially expressed genes in FLT3-ITD+/NPM1+ samples in all analyzed FAB types. The manuscript was well written but currently has a number of weaknesses that should be addressed.

Comments:

  • Is there any different splicing pattern in high and low allelic ratio of FLT3 in this cohort?
  • The authors stated “Hierarchical clustering did not reveal a specific cluster of FLT3-ITD+/NPM1+ samples suggesting limited specificity of the identified splicing events for this subgroup,” and described in line 189-191 “Furthermore, the splicing profiles of the double mutated samples were stronger as compared to either FLT3-ITD or NPM1” Can the authors explain what is the stronger splicing profile?
  • Only 33.3% differentially spliced genes could be validated in an independent M1 cohort (P value <0.2), 21.7% in M2 cohort, and 28.7% in M4 cohort. It seems the external validation was not very robust. Further, could the results be validated in the TCGA cohort?
  • I am wondering is there any clinical implication (such as treatment response or prognostic relevance) of the different alternative splicing patterns?
  • Is there any mutation distribution difference between the patients with NPM1+/FLT3-ITD+ and NPM1-/FLT3-ITD- in M1, M2, and M4 subgroups?
  • The authors showed 12 overlapped differentially spliced genes. Are there any mechanistic association between these genes and concurrent NPM1 mutation and FLT3/ITD?
  • The authors concluded that although FAB classification is not of prognostic use, maturation state-specific differential splicing of genes involved in regulation of DNA damage, and survival signaling could potentially affect the response of cells to (chemo- and targeted) therapy. However, I could not find the analysis in this study could support this assumption. Similarly, there was no functional study to suggest “these genes could serve as candidates for cell type-specific therapeutic targets, including EZH2 in FLT3-ITD+/NPM1+ M1 samples, Notch signaling in M2 patients and PI3K/AKT signaling or MELK kinase in M4 subtype. The authors should tone down the statement in “Conclusion”
  • Please clarify if P values in Figure 5 was correct (every P value in Figure 5 is 0.08?)
  • There was no figure legend for Figure 1D (not well illustrated).
